

# An analysis of the perceived causes leading to task-failure in resistance-exercises

Aviv Emanuel[1,2,3], Isaac Isur Rozen Smukas[2,3] and Israel Halperin[2,3]

[1] School of Psychological Sciences, Tel Aviv University, Tel Aviv, Israel
[2] School of Public Health, Sackler Faculty of Medicine, Tel Aviv University, Tel Aviv, Israel
[3] Sylvan Adams Sports Institute, Tel Aviv University, Tel Aviv, Israel

## ABSTRACT

**Background**. While reaching task-failure in resistance-exercises is a topic that attracts scientific and applied interest, the underlying perceived reasons leading to task-failure remain underexplored. Here, we examined the reasons subjects attribute to task-failure as they performed resistance-exercises using different loads.

**Methods**. Twenty-two resistance-trained subjects (11-females) completed one Repetition-Maximum (RM) tests in the barbell squat and bench-press. Then, in the next two counterbalanced sessions, subjects performed two sets to task-failure in both exercises, using either 70% or 83% of 1RM. Approximately 30 seconds after set-completion, subjects verbally reported the reasons they perceived to have caused them to reach task-failure. Their answers were recorded, transcribed, and thematically analyzed. The differences between the frequencies of the identified categories were then tested using a mixed logistic regression model.

**Results**. The most commonly reported reason was muscle fatigue (54%, $p < 0.001$), mostly of the target muscles involved in each exercise. However, remote muscles involved to a lesser extent in each exercise were also reported. Approximately half of the remaining reasons included general fatigue (26%), pain (12%), cardiovascular strain (11%), and negative affect (10%), with the latter three reported more often in the squat ($p = 0.022$).

**Conclusions**. In contrast to our expectations, task-failure was perceived to be caused by a range of limiting factors other than fatigue of the target muscles. It now remains to be established whether different perceived limiting factors of resistance-exercises lead to different adaptations, such as muscular strength and hypertrophy.

# INTRODUCTION

Whether one should reach task-failure in resistance-exercises is a question that attracts scientific and applied interest (*Nóbrega & Libardi, 2016*; *Davies et al., 2016*; *Sampson & Groeller, 2016*). Here we refer to task-failure as either momentary-failure (MF), the point in which an attempted repetition cannot be completed with proper form, or repetition maximum (RM), the final repetition one can complete prior to reaching MF (*Steele et al., 2017*). It has been established that reaching or approaching task-failure is important for

Corresponding author
Israel Halperin,
ihalperin@tauex.tau.ac.il

hypertrophy and strength development (*Davies et al., 2016*; *Morton et al., 2016*). Yet, to date, little is known about the subjective reasons underpinning task-failure. Although it can be expected that the inability to generate the required force with the target muscles is the main reason for task-failure, perception of fatigue, negative feelings, cardiovascular strain and pain can also be at play. Reaching task-failure due to one or more of the aforementioned factors could lead to different adaptations. For example, reaching task-failure because one cannot generate enough force with the target muscles is possibly more effective for hypertrophy purposes compared to reaching task-failure because of cardiovascular strain, or pain in body parts other than the target muscle groups. While investigating the underlying physiological causes of task-failure is a challenging task, examining the subjective aspects believed to cause task-failure is attainable and can shed light on this important issue.

One effective way to study subjective experiences during resistance training is with single-item scales (*Buckley & Borg, 2011*; *Helms et al., 2016*; *Hackett et al., 2017*). For example, rating of perceived effort (RPE) scales, such as Borg CR-10 (e.g., *Buckley & Borg, 2011*), can assist quantifying how much effort one is investing during, or after set completion, and allow for comparisons between different exercises, loads, and body parts involved in the exercise (*Duncan, Al-Nakeeb & Scurr, 2006*; *Buckley & Borg, 2011*). Interestingly, when sets are taken to task-failure, perceived effort is not always rated as maximal, indicating that effort is not the limiting factor in such cases (*Pritchett et al., 2009*). Such results led investigators to examine if other constructs can be the reason people terminate sets (*Steele et al., 2016*). A number of studies found that perception of discomfort, rather than RPE, was maximal at the point of set-termination, suggesting that discomfort can be a limiting factor (*Fisher & Steele, 2017*; *Stuart et al., 2018*).

Other popular effort scales gauge how many repetitions trainees estimate they can complete before reaching task-failure (*Helms et al., 2016*; *Hackett et al., 2017*). To illustrate, a rating of two indicates that only two more repetitions are left before reaching task-failure (*Hackett et al., 2017*). These scales are similar to RPE scales although developed solely for resistance training (*Helms et al., 2016*; *Hackett et al., 2017*). While single-item scales are practically useful and lead to theoretical insights (*Halperin & Emanuel, 2019*), they lack the resolution required to pinpoint the reasons why task-failure occurs. This is because they depend on a single question, which revolves around a single construct, to which people can only provide a single answer. Hence, other strategies of investigations are warranted.

Another way to investigate the causes leading to task-failure is to ask subjects why they terminated a set, and allow them to answer in an unrestrictive manner. This measurement strategy can expand our understanding of set termination that goes beyond the insightful, yet limited knowledge gathered via single-item scales. Such knowledge could lead to new research avenues. The limiting factors may vary as a function of the exercises completed and the loads lifted. Hence, the purpose of the present work was to examine the perceived causes trainees attribute to task-failure across different loads and exercises. To achieve this goal, we instructed resistance-trained subjects to reach task-failure in the squat and bench-press exercises under two load conditions (70% and 83% of 1RM). Within 20-40 s after set completion, we asked them what was the limiting factor in the set, and why they

did not perform another repetition. Subjects' answers were recorded, transcribed and analyzed.

## MATERIALS & METHODS

### Study design

The study consisted of three testing days with three to eight days between sessions. The first day consisted of 1RM tests for the barbell back-squat followed by the barbell bench-press, and an explanation of the experimental procedure. In the following two sessions, subjects performed two sets with the barbell squat followed by two sets with the barbell bench-press to task-failure, while lifting either 70%1RM in one session or 83%1RM in the other, in a randomized, counterbalanced order. While subjects were asked to reach MF in all sets, they were informed that they can terminate the set at RM. Hence, across days, task-failure was determined either by (1) inability to complete a repetition (MF), (2) subjects' decision to terminate the set based on their assumption that they can't complete another repetition (RM), or (3) technical failure determined by the experimenter.

The 70% and 83% of 1RM loads were selected for two reasons. First, they are within the recommended range for development of hypertrophy and strength for trainees with a range of training backgrounds (*Kraemer & Ratamess, 2004*). Second, based on pilot work and a previous study from our lab (*Emanuel, Rozen Smukas & Halperin, 2020a*), the two loads were expected to lead to considerable differences in the number of repetitions subject complete, yet lifting loads heavier than 83%1RM could have led subjects to perform as little as two or three repetitions in the final sets –an outcome we were preferred to avoid. Note that this this study is based on the protocol used in *Emanuel, Rozen Smukas & Halperin (2020b)* but addressed outcomes which were previously unexamined.

In the two experimental sessions, after each set, the researcher noted if the set was terminated due to RM or MF. The researcher then asked the subjects what were the limiting factors in the set, and why they could not perform another repetition. Subjects answered this question as they saw fit, without any restrictions. All answers were recorded with a tie-on microphone attached to the subjects' shirt and were later transcribed and analyzed. Subjects were asked to refrain from an intense training session 24 h prior to testing days and to avoid a heavy meal and caffeinated drinks or supplements at least three hours before testing sessions. All sessions were performed in the same facilities and ran by the same experimenter at approximately the same hour of the day ($\pm2$ h). No individuals other than the same single experimenter and a single subject were allowed to enter the lab during the experimental sessions.

### Subjects

Twenty-two resistance trained subjects volunteered to participate in this study (Table 1). Due to the exploratory nature of this work, a power analysis was not conducted. Yet, we assumed that subjects would perform at least 150 sets, which would provide us with an adequate amount to data to explore the topic at hand. Inclusion criteria consisted of healthy subjects between the ages of 18 and 45; a bench-press 1RM of at least 1.2 and 0.7 times the bodyweight for men and women, respectively; and at least 1.2, and 1 times the bodyweight

**Table 1  General demographics.**

| | Females ($n = 11$) Mean ± SD (Range) | Males ($n = 11$) Mean ± SD (Range) |
|---|---|---|
| Age | 29 ± 4 (23–38) | 30 ± 4 (22–37) |
| Height (cm) | 167 ± 6 (156–175) | 175 ± 6 (167–185) |
| Weight (kg) | 62 ± 7 (52–75) | 78 ± 4 (71.7–85.7) |
| Experience in RT (yrs) | 3 ± 2 (1–8) | 9 ± 4 (3–18) |
| Mean workouts per week | 3 ± 1 (1–5) | 3.5 ±.8 (2–5) |
| 1RM barbell bench press (kg) | 45 ± 9 (31–60) | 100 ± 14 (75–130) |
| 1RM/ Bodyweight bench press | 0.71 ± 0.12 (0.53–0.87) | 1.29 ± 0.22 (1–1.7) |
| 1RM barbell squat (kg) | 74 ± 14 (55–100) | 126 ± 20 (100–155) |
| 1RM/ Bodyweight squat | 1.19 ± 0.24 (0.81–1.50) | 1.6 ± 0.27 (1.2–1.6) |

in the squat. Subjects had to have at least one year of resistance training experience, and specifically at performing the free weight squat and bench-press. Additionally, subjects had to have some familiarity with taking sets to task-failure. Each subject signed an informed consent on the first day. This study was approved by the Tel-Aviv University institutional review 2019-0325.

## Procedures

**1RM tests (day 1).** Subjects were first weighed (Xinfu Household Electronics, Guandong, China) indicated their height, age and experience in strength training. They were then briefed on the study's aims, namely, to measure different perceptions associated with performance of the bench press and squat exercises to task-failure, across two different loads. All subjects then performed a squat to a height adjustable box which was set to achieve a knee angle of 115–120 degrees measured with a goniometer (mean knee angle = 118, SD = 5.93). Subjects had to lightly touch the box with their gluteus prior to initiating the concentric phase. During the bench-press subjects' preferred grip and body position were recorded and maintained throughout the study. In each repetition the bar must have lightly touched subjects' chest prior to the concentric phase. Subjects then performed a structured warmup protocol consisting of calisthenics and dynamic warmup followed by an individualized five-minutes warmup. This warmup protocol was identical in all sessions. Subjects then performed the barbell squat and bench-press 1RM protocol which consisted of a similar progression towards an estimated 1RM indicated by the subjects: 10,5,3,3,2,1 repetitions with empty bar, 40%, 60%, 70%, 80% and 90% of approximated 1RM, respectively. The increase in weight to the true 1RM was decided by the subjects and experimenter with 3–5 min of rest between 1RM attempts.

**Experimental sessions (day 2–3).** Following the general warmup protocol (see above), subjects performed the following specific warmup for the squat and again for the bench-press following the sets to task-failure with the squat: 10,5,3,1 repetitions with an empty bar with 40%, 55%, 70% of 1RM in the lighter day, or added another set of one repetition with 83% of 1RM in the heavier day. Following the last warmup set, subjects rested for two minutes and performed two sets to task-failure in the squat followed by the bench

press with either 70% or 83% of 1RM. Six minutes of rest were provided between sets and exercises. Subjects were instructed to perform the concentric portion of the lift as fast as possible, while maintaining a controlled ~2 s descend, as was assessed and insured by the experimenter. Within 20-40 s after set completion, subjects were asked what were the limiting factors in the set, and why they could not perform another repetition. The 20–40 s wait ensured subjects could catch their breath and sit down before providing an answer.

## Data preparation

We followed a similar approach used by *Halperin et al. (2016)*. Initially, general categories that were expected to account for set termination were extensively discussed by the authors. These categories included general and specific muscle fatigue, pain, negative affect, and cardiovascular strain (see below). Then, the first and last authors read all of the transcribed statements and examined if the agreed upon categories were present, needed to be refined, and whether other ones were noticed. In case that newer categories were identified, a discussion took place in order to decide if they should be included. Once the categories were agreed upon, the first and last authors picked at random a few statements and rated them simultaneously to confirm comparable ratings. Thereafter, the raters rated all statements individually. Each statement was rated once in a binary manner in each category. The raters then compared their overall ratings. Cases of mismatches were thoroughly discussed between the two raters until reaching an agreement. Note that the same statement could have been rated in more than one category. The final categories included the following:

*General fatigue*: statements with terms such as fatigue, tired, lack of energy, power or strength, all in relation to the whole body, or described as a general perception.

*Specific fatigue:* as described above, but the perceptions had to be attributed to muscles or a location in the body.

*Cardiovascular strain:* statements indicating that breathlessness or heartrate were the reasons for set-termination.

*Pain:* statements indicating a painful experience, including terms such as pain, hurt, pinch, and burning.

*Negative affect:* statements indicating an overall bad experience including terms such as annoyance, bad, terrible, and not fun.

Table 2 provides examples of the statements provided in each of the five categories across exercises, loads, RM/MF and gender, and their rating in each category. Since subjects provided the answers in Hebrew, the first author, who is fluent in both Hebrew and English, translated the statements into English.

## Data analysis

We tested for differences between the frequencies of the categories across loads and exercises in a mixed logistic regression model (the cardiovascular category was coded as 0). We next tested for differences in the frequency of each category between conditions and exercises via five separate mixed logistic regression models, one for each category (the frequency in a given category was the dependent variable while condition and exercise were the

Table 2 **Examples of answers provided to the question posed after set-completion: "What were the limiting factors of the set and why couldn't you perform another repetition?".** The ratings of each answer are divided by category, exercise, load, and set endpoint condition (* indicates MF).

| Load | Answers | Muscle Fatigue | General Fatigue | Affect | Cardio | Pain |
|------|---------|:--------------:|:---------------:|:------:|:------:|:----:|
| | | Squats | | | | |
| 70% | A general feeling of exhaustion. | | x | | | |
| 70%* | My quadriceps and hamstring muscles. | x | | | | |
| 70% | My lower back and my left quadriceps but also an unpleasant feeling in my body. | x | | x | | |
| 70% | I didn't reach failure, I just had enough. | | | x | | |
| 83% | I couldn't push with my legs any more, I feel fatigued. | x | x | | | |
| 83% | I felt stuck and a pressure in my lower back muscles. | | x | | | x |
| 83% | Mostly because of cardiovascular reasons but my legs were also hurting and the bar felt heavy on my shoulders. | | | | x | x |
| 83% | A general lousy feeling. I just wanted to get it over with. | | | x | | |
| 83% | A general feeling of fatigue of the whole system rather than just the muscles in my legs. | | x | | | |
| 83% | I run out of power and knew I wouldn't be able to complete another repetition. | | x | | | |
| | | Bench-press | | | | |
| 70% | The limiting factor was fatigue in my left shoulder muscles. | x | | | | |
| 70%* | I never felt my chest muscles fatigued like this before. | x | | | | |
| 70%* | I can't really explain it, I felt that I just ran out of strength. | | x | | | |
| 70% | My heartrate. I felt it from the very first repetition. | | | | x | |
| 83%* | Without a doubt it was my chest muscles. Not my triceps as I expected. | x | | | | |
| 83% | The limiting factor was mostly psychological. As if I gave up on the next repetition. | | | x | | |
| 83% | Pain in my left wrist. | | | | | x |
| 83% | I ran out of strength in my chest muscles. | x | | | | |
| 83%* | A combination of different reasons, including lower back pain, fatigue in my shoulder and arm muscles, and a feeling of breathlessness. | x | | | x | x |
| 83% | The limiting factor was my ability to produce strength with my upper body. | x | | | | |

independent variables). In all mixed regression models, random effects and interactions were added based on improvement in model fit, as indicated by the deviance statistic. In addition, the specific body parts mentioned, if any, under the muscle fatigue and pain categories were mapped (e.g., upper-body, quadriceps), counted, and reported. We also counted the frequencies of each category separately for males and females for exploratory purposes. Significance was set at $p < 0.05$. Statistical analyses and figures were carried out in R (version 3.6.0, R Core Team, Vienna, Austria) using the lme4 (*Bates et al., 2015*), cowplot (*Wilke, 2017*), and ggplot2 (*Wickham, 2011*) packages.

## RESULTS

Overall, 158 statements from terminated sets were recorded with only 37 sets terminated due to MF and the rest due to RM. We had ten missing observations in the 83%1RM

bench-press condition; four due to drop-outs and six due to technical difficulties. We had four more missing observations across the other conditions and exercises due to technical difficulties. The mean number of repetitions completed across sets and gender in the 70%1RM condition was $16.4 \pm 8.9$ (range: 6–55) and $13 \pm 2.9$ (range: 7–19) in the squat and bench-press respectively, whereas in the 83%1RM it was $8.1 \pm 3.3$ (range: 3–17) and $7.1 \pm 1.8$ (range: 4–12) in the squat and bench-press respectively.

**Overall frequencies of each category.** The final mixed logistic regression model included a random intercept, and its fixed effects were defined by the following equation:

$$Frequency_{ij} \sim \text{logistic}(Category_{ij} + u_{0j})$$

Out of the 158 statements, muscle fatigue was found to be the most frequent category accounting for 53.8% of the terminated sets ($OR = 9, b = 2.2, SE = 0.29, z = 7.42, p < 0.001$, 95% CI [1.8–2.9]). The second most frequent category was general fatigue accounting for 25.9% of the terminated sets ($OR = 2.7, b = 1, SE = 0.31, z = 3.24, p = 0.001$, 95% CI [0.45–1.67]), followed by pain (12%), cardiovascular (11.4%) and negative affect (10.1%). No significant differences were found between the frequency of either pain ($OR = 1, b = 0.06, SE = 0.35, z = 0.17, p = 0.861$, 95%CI [−0.66–0.74]) or negative affect ($OR = 0.87, b = −0.13, SE = 0.36, z = −0.36, p = 0.716$, 95%CI [−0.87–0.59]) and the cardiovascular category (See Fig. 1).

**Differences in frequencies between exercises and loads, males and females.** The results of the mixed logistic regression models tested for each separate category are presented at Table 3. The frequencies the categories by gender are reported in Table 4.

## DISCUSSION

Here we examined what limiting factors were perceived to lead to task-failure in the squat and bench-press exercises using 70% and 83% of 1RM, among resistance-trained subjects. Subjects were required to state what were the limiting factors of each set within 20–40 s after reaching task-failure. Of the five categories used to map subjects' responses, specific muscle fatigue was the most frequently reported. Within the muscle fatigue category, the target muscles (i.e., prime movers) in each exercise were specified in most, but not all of the responses. The remaining responses were categorized as general fatigue, pain, negative affect and cardiovascular strain. Some differences were observed between exercises with task-failure in the squat being attributed more often to negative affect and, while statistically insignificant, in cardiovascular strain compared to the bench-press exercise. Similarity, some differences were observed between loads, in which lighter loads led to greater pain and cardiovascular strain compared to heavier loads. No differences in the distribution of categories were identified between males and females.

As can be expected, most sets were perceived to be terminated because of muscle fatigue attributed to target muscle groups (i.e., lower body for squats and upper body for bench-press). However, within this category, certain variability was noted. In the squat, 27% (13/47) of the statements within the muscle fatigue category were attributed to the upper body and lower back, indicating that in these cases, task-failure was not perceived to be due to the target muscles. Moreover, different muscles within the legs

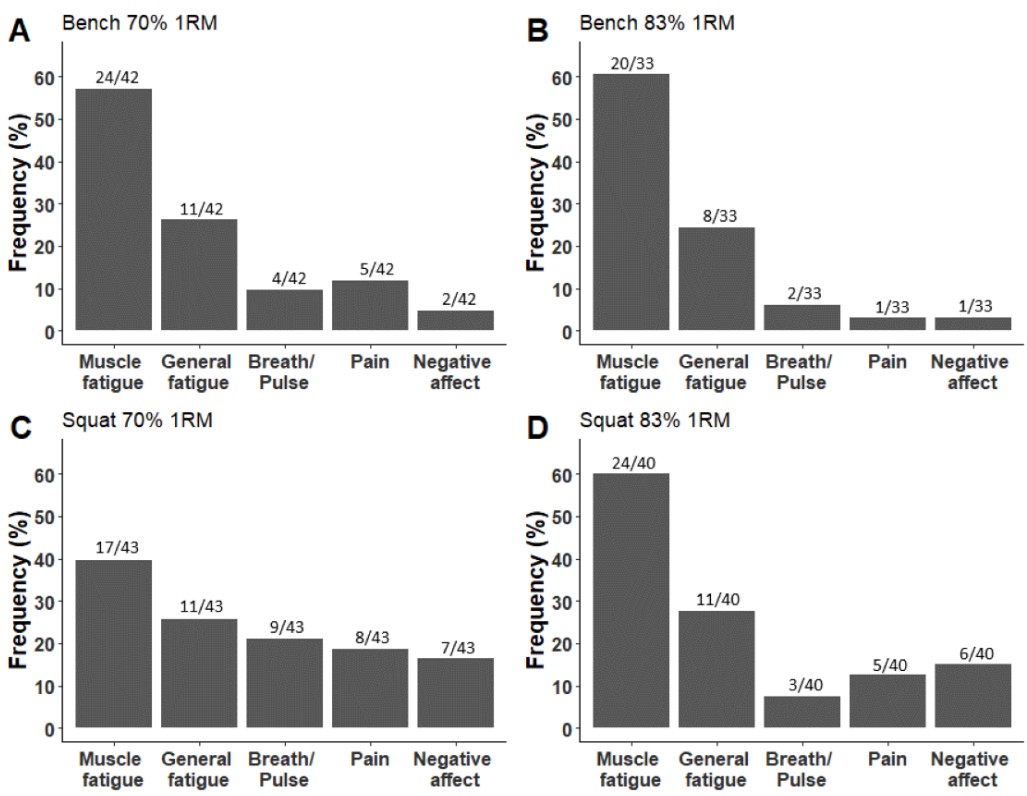

**Figure 1** **The distribution of each category by exercises and loads.** (A–E) depict the muscle fatigue, general fatigue, pain, cardiovascular, and negative affect categories, respectively. The $y$-axis represents the percent of each category, calculated as the number of ratings, divided by the number of sets in each exercise and load.

were mentioned, including the gluteus, quadriceps and hamstring, indicating that limiting factors within the target muscle groups can vary between individuals. In the bench-press, some variability was also noted within the target muscle groups, with the chest muscles being reported the most, followed by the arms (presumed to be triceps) and then the shoulders. The variability in reporting which of the target muscle groups was the limiting factor in each exercise can partly be explained by technical variation in the way the exercises were performed. For example, different stance widths, foot placement, and forward lean during the squat can affect the moments at the hip and knee joints (*Escamilla et al., 2001*; *Glassbrook et al., 2017*; *Lorenzetti et al., 2018*). It is interesting to consider whether variations in the perceived muscle group being the limiting factor in a given exercise can lead to different adaptations, such as strength and hypertrophy. Alternatively, whether suboptimal adaptations may occur when unrelated muscles are considered to be the limiting factor in exercise-performance.

General fatigue was also frequently attributed to set termination across exercise and loads. This could indicate that multi-joint exercises involving large muscle groups produce a global and unspecified feeling of fatigue strong enough to lead to set-termination. While pain, negative affect, and cardiovascular strain were mentioned fewer times compared to

Emanuel et al. (2020), *PeerJ*, DOI 10.7717/peerj.9611

**Table 3** **Mixed model logistic regression results.**

| Model | Random terms | Fixed terms | OR | Estimate (b) | SE | z | p-value | 95% CI (LL, UL) |
|---|---|---|---|---|---|---|---|---|
| $Pain_{ij} \sim$logistic $(b_{0ij} + b_1{}^*Squat\_Exercise_{ij} + b_2{}^*83\%1RM\_Condition_{ij+}\ u_{0j})$ | $b_{0j} = \gamma_{00} + u_{0j}$ $b_1 = \gamma_{10} + u_{1j}$ | $83\%1RM\_Condition_{ij}$ | 0.16 | −1.81 | 0.93 | −1.92 | 0.053 | −12.43, 0.01 |
| | | $Squat\_Exercise_{ij}$ | 4.74 | 1.55 | 3.59 | 0.43 | 0.665 | −6.87, 7.38 |
| $Negative\_affect_{ij} \sim$logistic $(b_{0ij} + b_1{}^*Squat\_Exercise_{ij} + b_2{}^*83\%1RM\_Condition_{ij+}\ u_{0j})$ | $b_{0j} = \gamma_{00} + u_{0j}$ | $83\%1RM\_Condition_{ij}$ | 0.85 | −0.16 | 0.55 | −0.29 | 0.770 | −1.44, 1.05 |
| | | $Squat\_Exercise_{ij}$ | 4.59 | 1.52 | 3.59 | 2.27 | 0.022 | 0.50, 16.11 |
| $General\_fatigue_{ij} \sim$logistic $(b_{0ij} + b_1{}^*Squat\_Exercise_{ij} + b_2{}^*83\%1RM\_Condition_{ij+}\ u_{0j})$ | $b_{0j} = \gamma_{00} + u_{0j}$ | $83\%1RM\_Condition_{ij}$ | 0.99 | −0.01 | 0.38 | −0.04 | 0.967 | −0.85, 0.81 |
| | | $Squat\_Exercise_{ij}$ | 1.06 | 0.06 | 0.38 | 0.17 | 0.860 | −0.71, 0.92 |
| $Muscle\_fatigue_{ij} \sim$logistic $(b_{0ij} + b_1{}^*Squat\_Exercise_{ij} + b_2{}^*83\%1RM\_Condition_{ij+}\ u_{0j})$ | $b_{0j} = \gamma_{00} + u_{0j}$ | $83\%1RM\_Condition_{ij}$ | 1.84 | 0.61 | 0.35 | 1.72 | 0.084 | −0.05, 1.38 |
| | | $Squat\_Exercise_{ij}$ | 0.63 | −0.45 | 0.35 | −1.30 | 0.192 | −1.19, 0.26 |
| $Cardiovascular_{ij} \sim$logistic $(b_{0ij} + b_1{}^*Squat\_Exercise_{ij} + b_2{}^*83\%1RM\_Condition_{ij+}\ u_{0j})$ | $b_{0j} = \gamma_{00} + u_{0j}$ | $83\%1RM\_Condition_{ij}$ | 0.23 | −1.44 | 0.75 | −1.91 | 0.056 | −10.80, −0.001 |
| | | $Squat\_Exercise_{ij}$ | 3.29 | 1.19 | 0.70 | 1.69 | 0.090 | −0.11, 5.03 |

**Notes.**

OR, odds ratio; SE, standard error; CI, confidence interval; LL, lower limit; UL, upper limit.

**Table 4 Frequency (percent) of each category by gender and set endpoint condition.**

|  |  | Muscle fatigue | General fatigue | Negative affect | Pain | Cardio |
|---|---|---|---|---|---|---|
| Gender | Females | 42/80 (52.5%) | 19/80 (23.8%) | 7/80 (8.7%) | 10/80 (12.5%) | 12/80 (15%) |
|  | Males | 42/76 (55.3%) | 22/76 (28.9%) | 8/76 (10.5%) | 8/76 (10.5%) | 6/76 (7.9%) |
| Set end-point | MF | 25/37 (67.5%) | 8/37 (21.6%) | 0/37 (0%) | 4/37 (10.8%) | 2/37 (5.4%) |
|  | RM | 60/121 (49.6%) | 33/121 (27.3%) | 16/121 (13.2%) | 15/121 (12.4%) | 16/121 (13.2%) |

specific and general fatigue, some interesting patterns emerged. Mainly, the results indicate that more sets in the squats were terminated due to pain and negative affect compared to bench-presses, and that in the 70%1RM squat condition more sets were terminated due to cardiovascular strain. Some of these findings are aligned with other studies. For example, lower body exercises have shown to cause greater degree of negative affect (*Portugal et al., 2015*), and require greater energy expenditure (*Lyons et al., 2007*; *Flanagan et al., 2014*; *Vianna et al., 2014*) compared to upper body exercises. These findings suggest that squats may be limited by a wider range of factors, compared to the bench-press. As a whole, these results are also alighted with other studies reporting that other constructs, such as discomfort, can be a limiting factor during sets (*Steele et al., 2016*).

This study has several methodological aspects worthy of discussion. First, asking subjects to answer the question concerning set termination within ~30 s can be viewed both as a strength and a limitation. Asking the question in proximity to set termination was expected to lead to a more accurate answer, but given the physically challenging nature of the task, subjects may have provided less details. Similarly, subjects were allowed to answer the question without any restrictions, but this may have caused important information to be lost that a more structured questioning procedures might have captured. Second, only 23% of sets were completed due to TF, despite subjects being encouraged to reach TF in all sets. It is possible that set termination due to RM and TF lead to different perceptions of its underlying causes. However, since it is ethically impossible to enforce TF, overcoming this limitation is a challenging task. Third, while subjects in this study were experienced in RT, their unique training background could have shaped their responses. Moreover, subjects' beliefs and attitudes about the factors leading to set termination, as well as their general knowledge pertaining to resistance training may have also played a role in their responses (e.g., *Vaegter et al., 2020*). Third, no physiological outcome, such as heartrate, was measured and correlated with participants' statements. In view of the aforementioned points, future studies should collect more data on trainees' training background and knowledge, attitudes and perception using structured or semi-structured interviews and also collect physiological measures.

As studies continue to investigate how many repetitions one completes relative to task-failure, it may be of added value to examine what are the limiting factors leading to set-termination. This is because different limiting factors within sets could lead to different acute and long-term adaptations that might not be aligned with the sought-after outcomes. Investigating limiting factors can be done by directly measuring the possible physiological

pathways leading to set-termination. Such studies require complex designs and equipment. An alternative is to study the subjective experiences accounting for set-termination, as was done in the present study, and to our knowledge, for the first time. The present investigation may indicate that relying on task-failure as a standard for load and repetition prescription may fail to capture a variety of limiting factors other than fatigue of the target muscles. Thus, it could be that in addition to prescribing one to reach task-failure, or proximity to task-failure, monitoring other aspects of set-termination may prove beneficial. For example, loads can be modified in cases that cardiovascular strain or negative affect are reported at set-termination, and exercise technique can be modified in cases that target muscles are not perceived to be a limiting factor. However, it is required to first establish what type of relationships, if any, exist between perception of the limiting factors and actual adaptations.

## CONCLUSIONS

We observed that set-termination was mostly perceived by subjects to be a result of muscle fatigue in the target muscles, followed by a general feeling of fatigue, negative affect, pain and cardiovascular causes. These reasons were found to vary between exercises and loads. The results of this study show that there are a variety of perceived reasons for set termination, which might affect exercise-adaptations, and warrant further investigation.

### Funding
The authors received no funding for this work.

### Competing Interests
The authors declare there are no competing interests.

### Author Contributions
- Aviv Emanuel and Israel Halperin conceived and designed the experiments, analyzed the data, prepared figures and/or tables, authored or reviewed drafts of the paper, and approved the final draft.
- Isaac Isur Rozen Smukas performed the experiments, authored or reviewed drafts of the paper, and approved the final draft.

### Human Ethics
The following information was supplied relating to ethical approvals (i.e., approving body and any reference numbers):
Tel Aviv University Institutional Review Board approved this study (2019-0325).

### Data Deposition
The raw data are available as a Supplemental File.

## Supplemental Information

Supplemental information for this article can be found online at http://dx.doi.org/10.7717/peerj.9611#supplemental-information.

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
