# Peer review of "An analysis of the perceived causes leading to task-failure in resistance-exercises"

_PeerJ, doi:10.7717/peerj.9611_

## Round 0.1 · original submission · Major Revisions

Thanks for your submission. This is a well written manuscript in an important area of research. Please respond to each of the reviewers' comments and suggestions and be sure to address them adequately. I look forward to reviewing the next iteration. Scotty

·

Basic reporting

The authors' reporting of the data was clear and effective. The manuscript is very well written in its current form.

Experimental design

The study's experimental design was appropriate given the authors' research question. The study does not have any major or fatal flaws that prohibit it from being considered for publication. The approach to the problem was straightforward and easy to understand.

Validity of the findings

The findings of the study were valid given the research question and design.

Additional comments

Within the last decade, the need to perform resistance training exercises to failure has drawn attention from researchers and practitioners, with some evidence demonstrating that doing so enhances outcomes related to muscle hypertrophy. However, the perceptual aspect of training to failure and why trainees can't complete additional repetitions have received little attention in the literature. As such, the authors of the present study have attempted to understand these perceptions by asking participants their reasons for discontinuing a set.

I found this research study to be interesting, with the topic being worthy of additional attention from the field. The general premise of the research question was fairly simple, yet some aspects of the findings were surprising. The manuscript is very well-written and the study's design does not feature any fatal flaws that prevent this work from eventually being accepted. With this being stated, I have a few comments for the authors that need clarity during subsequent revisions. I have separated my thoughts into major and minor comments, and have generally tried to list them in chronological order. Overall, a job well done by the authors.

MAJOR COMMENTS
-One thing I believe worth mentioning further is the fact that all of the participants in this study were resistance trained. It would seem that each individual's unique resistance training experience, including styles of training, frequency, etc. would shape how one perceives training until failure. Did the authors capture any data pertaining to whether or not these participants regularly trained to failure, as well which exercises they train to failure with?

-The beliefs and attitudes about muscle fatigue, as well as general knowledge and previous education pertaining to resistance training anatomy are also probably very important. I'd imagine that each participant had his/her own perceptions about this topic prior to even engaging in the exercise. Was this considered?

-Though the study's sample size was relatively small, I think exploration of sex differences is worthy of consideration. It would be interesting to learn if the perceptual responses differ for men versus women. If so, this may have implications for sex-specific resistance training guidelines.

-Similar to the previous two comments, it would be interesting for future studies to explore whether these perceptual responses change as individuals progress in their training (i.e., going from a novice to mid-level trainee).

-Please clarify that each participant performed these exercises under controlled conditions and not with other participants nearby.

-Personally I don't care for Figure 1, as I think it makes more sense to display each of the five categories on the same graph(s). Maybe one approach would be to create a graph for each condition (i.e., bench and squat at 70% and 83% 1RM), with the percentage of the selected areas shown. Another simple solution might be to use pie charts.

MINOR COMMENTS
-Please consider incorporating the number of repetitions performed into the analysis, as there may be wide variability despite a relative % of the 1RM selected. Perhaps there is a trend with respect to certain perceptual responses and the number of repetitions performed? For example, those that performed more repetitions may have had higher heart rates and therefore more complaints related to cardiovascular strain.

·

Basic reporting

Thank you for the opportunity to review your submission to PeerJ. The purpose of this manuscript was to investigate potential causes leading to task-failure in resistance-type exercises in trained men and women. With this submission, the authors attempt to provide insight to a unique question in the field of exercise science. However, I have some suggestions that I believe will improve the quality of the paper.

Experimental design

- Unless I am missing it, the paper seems to lack a classic “purpose statement”. I would recommend the authors adding more direction here, which will increase the impact of your work.

- The use of two experimental days is not clear. At later points in the manuscript the authors reveal a “heavy” and a “light” day; however, this is not clarified in the experimental design section. I recommend the authors consider the use of a schematic diagram to clearly present this to the reader. Further, the authors should detail what randomization procedure was placed on the “day”. I.e., was the heavy day always second? This may impact the outcomes of the paper.

Validity of the findings

- Within the data frame shared by the authors, there are a few things of note. Two subjects are highlighted without note as to why. Also, there are some variables listed in this data frame, that are not discussed in the manuscript (i.e. ,barbell velocity). I defer to the direction of the editor, but you may look into reporting those data within the manuscript or removing the variables from your data table. The additional variables that are not discussed may cause confusion to readers of you work
- I there a reason the subject numbers start with 3 vs 1?
- Subject 23 and 24 seem to have a bit of missing data – subject 24 has 0 data outside of demographics – is this the technical issues the authors mentioned?
- Small error in subject 19 weight
- Please consider placing units by each variable

Additional comments

Figures and Tables:
Figure 1: I appreciate the authors clearly presenting the frequency of each of their responses. However, I’m not sure this is the best way to describe your findings. Have the authors looked into tables with the outcomes of their statistical models vs. these bar charts?
Introduction
1. I believe the terms momentary-failure and repetition maximum may not be the most appropriate. Specifically, I believe the field recognizes “repetition maximum” as the maximum load which a person can lift maximally, vs. the final repetition one can complete prior to reaching MF (as used in the present work). I believe these definitions may lead to ambiguity, which may limit your impact. Please consider revising/clarifying.
2. Line 48: When describing the “reasons” underpinning task-failure, I recommend adding some clarity here. Are the authors referring to physiological or psychological? The preceding sentence suggests you are discussing “physiological reasons” which cause task failure, though the rest of the manuscript does not have this focus.

Methods
3. Was there an a priori power analysis performed?
4. Line 101: The reference listed here is not properly cited. Upon searching the internet, I have found a pre-print, but am unsure if this is what the authors are citing.
5. Line 122: Please update from “weighted” to “weighed”. Also, can the author provide additional detail on how this was performed (i.e., calibrated scale?). To confirm, height, age and training age was self-reported?
6. Line 123: Can the authors give me more detail on how the participants were “briefed on the study’s aims”? I am concerned that the subjects being aware of what you were looking at may have impacted your outcome measures.
7. Line 124-125: How was the knee angle assessed?
8. Line 143-144: How was a ~2 second descent insured for each repetition? Or was this a general observation? I believe either is fine, but adding clarity will increase reproducibility of your methods. Similarly, why did the authors wait 20-40 seconds after set completion to interview the subjects?
9. Lines 181-182: Can you please provide the specific examples of where random effects were added (and what they were specifically)? The same should be noted for where (and what) interactions were placed in the models. Both of these parameters may significantly impact the interpretations and should not be overlooked.
10. Please cite R appropriately. This can easily be done using the “citation()” argument within R itself.
11. I appreciate you recognizing your use of the “lattice” package (please cite appropriately using citation(“lattice”)), but I do not believe this package can run your mixed logistic regression models. If that is accurate, could you please list the relevant package(s) you used?
12. How were model assumptions checked?
Results
13. Only ~23% of statements received were due to momentary failure vs. repetition maximum lifts. Have the authors considered how this may impact their outcomes?
14. The authors report 10 missing observations in the heavier day – this leads me to believe these data were not missing at random and may be an issue to be investigated further. Have the authors considered this?
15. Due to the amount of missing observations, I would recommend a modified version of a CONSORT diagram to be presented. This may be a way to clearly represent the data collected, what is missing, and the final sample used for analyses.
16. Line 191: You describe muscle fatigue as being the most frequent category responsible for terminated sets. However, muscle fatigue was not directly described in your methods. Is this a special case of “specific fatigue (line 163)”?
17. I appreciate the reporting of the beta, SE and P values by the authors. However, have the authors considered reporting odds ratios instead of the coefficients?
18. The statistics reported in the results may be better listed in a separate table to allow the reader to quickly and easily compare between models.
Discussion
19. The authors do a nice job of summarizing their findings. However, there is minimal comparison of the findings presented in the present manuscript to the body of literature as a whole. For example, I believe there is only one reference in the discussion section in total. I recommend the authors consider significant revision to this section to better compare/contrast with the available literature.
20. How does the current population impact your results? I would argue that some of your subjects were not very well trained (based off of low training age and reported 1RM strength), while others may be particularly trained. Is it possible that the heterogeneous sample may have impacted your findings?

·

Basic reporting

The reporting within this manuscript is excellent. The writing is clear and unambiguous. The Introduction does a nice job of framing the research question and the need for this experiment. The Discussion is concise and provides meaningful interpretations for the reader. The only recommendation for the reporting is to include “0’s” prior to the decimal point (i.e., p = 0.716).

Experimental design

The experimental design used in this study is appropriate to address the well-articulated research question. The design is simple yet constructed in a way that faithfully answers the research question. The methods described are thorough and will allow for replication.

Validity of the findings

The experimental design of the study provides high external validity. The use of common exercises (e.g., squat and bench press) in a resistance trained population with commonly used exercise intensities allows the findings of this study to be generalized to this population. The author's speculation on possible differential adaptations due to different limiting factors provides a meaningful avenue for future research.

Additional comments

It was a pleasure to review this manuscript. I have a few minor suggestions that are intended to improve the quality of an already high-quality manuscript.
Line 48 would be strengthened if the authors include ‘subjective’ before ‘reasons underpinning task-failure’.
Line 81, the sentence that begins “Hence,...” would be improved if broken into two sentences.
For instance: Hence, in this study we instructed resistance-trained subjects to reach task-failure in the squat and bench-press exercises under two load conditions (70% and 83% of 1RM). After set completion, we asked them what was the limiting factor in the set and why they could not perform another repetition. Their answers were recorded...:”
Line 97, I appreciate and understand the author's description of their decision to use the intensity in the study (70 & 83%). Yet, it is still not entirely intuitive why 2 separate (somewhat) similar intensity zones were necessary for the experiment. I do not feel this takes away from the findings, but if some minor additional details could be provided here it would benefit the manuscript.
Line 122, weighed instead of weighted

---

## Round 0.2 · accepted · Accept

Thank you for your attention to detail in this revised version. I am happy to accept it as submitted. Thanks again for your article!

·

Basic reporting

No comment

Experimental design

No comment

Validity of the findings

No comment

Additional comments

At this point I have no further comments or concerns. I believe that this paper will make a solid contribution to the resistance training/perception literature. Well done!

·

Basic reporting

The reporting of data, previous literature and ideas are clear and well done.

Experimental design

The design of the study matches well with the proposed research topic.

Validity of the findings

All data have been provided and support the authors' findings.

Additional comments

I wish to thank the authors for their careful revisions. I very much appreciate all of the responses to my comments. I think the manuscript as a whole has improved and is great. I have just couple of very minor comments for your to consider moving forward:

Model Table: I like the addition of this table, as it allows the reader access to your models completely. I also appreciate the logical expression of your random effects. Though, I further recommend specifying your random effects in the text (e.g., random slope per participant, random intercepts and slopes per participant etc…) to further clarify for the reader who may incorrectly interpret the elements of your model within your table.

Dataset: There is a “Sheet1” tab that has a few random integers. Unless you have another purpose for this that I have missed, please remove this tab to avoid confusion for the readers. Within your dataframe itself, there are still two columns that are fully without data (i.e., height_saftey_bench, dip (m)) that you should also consider either filling in with data or to remove to make things clean for the reader. Lastly, cell U,6 has an empty comment from Isaac Rozen. The dataset otherwise looks easy to comprehend and is very much appreciated.

·

Basic reporting

The reporting is clear and unambiguous. The revised version is indeed improved.

Experimental design

The experimental design is sounds and well described. The revised version of the manuscript has improved the clarity of the experimental design.

Validity of the findings

The main findings are well articulated and the revised version of the manuscript has improved the overall quality and generalizability.

Additional comments

The authors have adequately addressed my comments. The manuscript is improved. Well done.